# Natural Language Processing for Information Extraction of Gastric Diseases and Its Application in Large-Scale Clinical Research

**DOI:** 10.3390/jcm11112967

**Published:** 2022-05-24

**Authors:** Gyuseon Song, Su Jin Chung, Ji Yeon Seo, Sun Young Yang, Eun Hyo Jin, Goh Eun Chung, Sung Ryul Shim, Soonok Sa, Moongi Simon Hong, Kang Hyun Kim, Eunchan Jang, Chae Won Lee, Jung Ho Bae, Hyun Wook Han

**Affiliations:** 1Department of Biomedical Informatics, CHA University School of Medicine, CHA University, Seongnam 13488, Korea; gkwlal@gmail.com (G.S.); sungryul.shim@gmail.com (S.R.S.); soonoksa19@gmail.com (S.S.); moongi.simon.hong@gmail.com (M.S.H.); anomii1@naver.com (K.H.K.); eunchan.j93@gmail.com (E.J.); cwl941021@gmail.com (C.W.L.); 2Institute for Biomedical Informatics, CHA University School of Medicine, CHA University, Seongnam 13488, Korea; 3Department of Internal Medicine and Healthcare Research Institute, Healthcare System Gangnam Center, Seoul National University Hospital, 39FL Gangnam Finance Center 152, Teheran-ro, Gangnam-gu, Seoul 06236, Korea; medjsj7@gmail.com (S.J.C.); suji421@gmail.com (J.Y.S.); syyang@snuh.org (S.Y.Y.); icetea@snuh.org (E.H.J.); gohwom@daum.net (G.E.C.); 4Department of Health and Medical Informatics, Kyungnam University College of Health Sciences, Changwon 51767, Korea

**Keywords:** endoscopy, digestive system, gastritis, natural language processing, information extraction

## Abstract

**Background and Aims**: The utility of clinical information from esophagogastroduodenoscopy (EGD) reports has been limited because of its unstructured narrative format. We developed a natural language processing (NLP) pipeline that automatically extracts information about gastric diseases from unstructured EGD reports and demonstrated its applicability in clinical research. **Methods:** An NLP pipeline was developed using 2000 EGD and associated pathology reports that were retrieved from a single healthcare center. The pipeline extracted clinical information, including the presence, location, and size, for 10 gastric diseases from the EGD reports. It was validated with 1000 EGD reports by evaluating sensitivity, positive predictive value (PPV), accuracy, and F1 score. The pipeline was applied to 248,966 EGD reports from 2010–2019 to identify patient demographics and clinical information for 10 gastric diseases. **Results:** For gastritis information extraction, we achieved an overall sensitivity, PPV, accuracy, and F1 score of 0.966, 0.972, 0.996, and 0.967, respectively. Other gastric diseases, such as ulcers, and neoplastic diseases achieved an overall sensitivity, PPV, accuracy, and F1 score of 0.975, 0.982, 0.999, and 0.978, respectively. The study of EGD data of over 10 years revealed the demographics of patients with gastric diseases by sex and age. In addition, the study identified the extent and locations of gastritis and other gastric diseases, respectively. **Conclusions:** We demonstrated the feasibility of the NLP pipeline providing an automated extraction of gastric disease information from EGD reports. Incorporating the pipeline can facilitate large-scale clinical research to better understand gastric diseases.

## 1. Introduction

### Background

Gastric cancer is the second most prominent cancer and the leading cause of cancer-related deaths in Korea, with a high prevalence of *Helicobacter pylori *(*H. Pylori*)** as the major environmental risk factor for the development of gastric cancer [1,2]. Population-based screening for gastric cancer was implemented in 2002 as part of the National Cancer Screening Program (NCSP) in Korea, and approximately 7 million esophagogastroduodenoscopies (EGDs) were performed in 2019 as part of the NCSP [3]. The national-level program generates millions of EGD reports generated from endoscopy procedures, which contain a range of information about diagnosis and specific phenotypes of gastric disease. However, the utility of this information has been limited. Due to the unstructured format of the reports and usage of endoscopic abbreviations, information extraction from EGD reports is dependent on a manual review by experts. Furthermore, the discordance between the pathologic reports and endoscopic procedures in electric health records makes data extraction even more challenging. This process has greatly limited the use of big data in endoscopy for quality tracking or research.

Natural language processing (NLP) offers automated extraction and structuring of information from unstructured text documents using a set of computational techniques [4]. Among the wide range of applications of NLP technology in the biomedical field, information extraction from electronic health records is one of the popular applications [5]. For example, extraction of biomedical terms and relations from texts using deep learning-based NLP models, such as BioBERT (Bidirectional Encoder Representation from Transformers for Biomedical Text Mining), recurrent neural network-based models, and Med7, are now increasingly researched [6,7,8]. In particular, NLP studies have been conducted in procedure-heavy sub-disciplines such as radiology and gastroenterology that generate procedural or operational reports [4,9,10,11,12,13,14,15]. Do et al. used the NLP system to gather patterns of metastatic spread from radiology reports in a large patient cohort [9]. Bae et al. developed an NLP system to extract information related to neoplasm from colonoscopy reports and evaluated the quality benchmarks of endoscopists [10]. However, NLP has not yet been used to extract information from EGD reports.

In this study, we developed an NLP pipeline to extract the key concepts, including extent, location, stage, and size, from EGD reports and their linked pathology reports for 10 gastric diseases. The performance of the pipeline was evaluated in comparison to the gold standard created by gastroenterologists. For demonstrating NLP’s utility in clinical research, the pipeline was applied to 10-years of EGD data from a health check-up center and was used to identify the demographics of gastric diseases.

## 2. Materials and Methods

### 2.1. Study Design and Setting

All EGD reports compiled during gastric cancer screening were identified from the Seoul National University Hospital (SNUH) Healthcare System, Gangnam Center, where comprehensive medical check-ups are conducted each year for approximately 30,000 patients. A total of 365,801 EGD reports and 73,537 linked pathology reports from 140,694 patients between 2003 and 2019 were extracted from the SNUH clinical data warehouse (Figure 1). A representative sample of 3000 EGD reports paired with pathology reports was randomly selected for the development dataset of the NLP pipeline. The development dataset was split into a train dataset (2000 reports) for pipeline design and a test dataset (1000 reports) for validation. This study was approved by the Institutional Review Board of SNUH (approval no. 1909-093-670) and conforms to the provisions of the Declaration of Helsinki (as revised in Fortaleza, Brazil, October 2013).

### 2.2. Target Variables for Information Extraction

Our aim for developing the NLP pipeline was to extract specific information for five common types of chronic gastritis and another five gastric diseases in Korea. Chronic gastritis includes three types of *H. pylori*-related gastritis (atrophic gastritis, intestinal metaplasia, and follicular gastritis), and two types of gastritis with high prevalence in Korea (superficial gastritis and erosive gastritis). The endoscopic description was based on Schindler’s classification. Other gastric diseases, including gastric ulcers, two types of polypoid lesions (polyp, submucosal tumor), and two types of neoplastic diseases (dysplasia and cancer), were also extracted by the NLP.

For gastritis, information on the presence and anatomical extent or degree of the lesion was extracted. The extent consisted of three levels: extent 1 indicated that gastritis was observed only in the antrum (lower part of the stomach), extent 2 indicated that gastritis was observed only in the body/fundus (mid to upper part of the stomach), and gastritis in both the antrum and body/fundus was defined as extent 3. For gastric ulcers and neoplastic diseases, the pipeline extracted their presence, anatomic location, and size of the lesion in centimeters from the EGD report. In the case of gastric ulcers, information on the stage of the disease consisting of active, healing, and scarring was also extracted. Information on gastric dysplasia and cancer was extracted from the combined EGD and pathology reports. The gastric dysplasia category included all levels of adenomas in the pathology report; adenocarcinoma, neuroendocrine tumor, and all types of lymphoma were included in the gastric cancer category.

### 2.3. NLP Pipeline Development

The NLP pipeline was developed using Python (3.7.10, Python Software Foundation) and the regular expression package ‘re’. Regular expressions are scripts specialized for processing texts, such as pattern matching and capturing terms [16]. The EGD reports included three types of language forms: Korean, English, and Korean with English terminology. Of the 3000 EGD report in the development dataset, 75 EGD reports were written in English only, and the rest of 2925 EGD reports were written in Korean with English terminology. Reports written in Korean only were not included in the development. Therefore, we needed to create an NLP pipeline that can process multi-language reports written in Korean, English, or both, and build a lexicon of Korean-English medical terms, synonyms, and endoscopic abbreviations (Appendix A). The customized NLP dictionary was based on a glossary in an endoscopy textbook, the Medical terminology, 6th edition, by the Korean Medical Association, and the terms in the train dataset [17,18]. According to the structure of the EGD report, the dictionary was divided into two categories: findings and impressions. The endoscopic findings section contained terms that describe the phenotypic characteristics of gastric diseases or abnormalities, and the impressions section contained the diagnostic terms used by the examiner during the EGD.

The NLP extracts gastric disease information in four steps: text preprocessing, concept mapping, concept extraction, and summarizing (Figure 2). In the text preprocessing step, the pipeline filters out information unrelated to gastric diseases, such as descriptions of the esophagus and duodenum, leaving only the text related to the stomach. Unexpected end-of-line characters and white spaces are removed from the sentence. In the concept mapping step, the pipeline refers to the dictionary to find key medical terms in the preprocessed text using the following sub-steps: when a diagnostic term is found in the impressions section of the EGD report, the pipeline maps key sentences containing terms describing the phenotypic nature of the diagnostic term in the endoscopic findings section of the EGD report. Concepts such as location and size are extracted from the mapped key sentences in the concept extraction step. Finally, the extracted concepts are summarized in a predefined format (Figure 3). The NLP pipeline was iteratively updated until performance improvement was possible through rule updates. The final version of the pipeline was validated using 1000 EGD reports (test dataset).

### 2.4. Application of the NLP Pipeline

The NLP pipeline was applied to 248,966 consecutive EGD reports and 50,096 associated pathology reports in 97,998 patients aged between 18 and 100 years at the SNUH Gangnam Center from January 2010 to December 2019. The pipeline extracted information about the demographics, anatomical extent, and location of 10 gastric diseases.

### 2.5. Statistical Analysis and Performance Evaluation

Continuous variables were calculated as mean values with standard deviation. Discrete data were tabulated as numbers and percentages. The chi-squared test was used to compare proportions, and the t-test was used to compare quantitative variables. The NLP pipeline was assessed by calculating the overall agreement between the pipeline and the gold standard created by two gastroenterologists with manual annotation. The performance of information extraction was evaluated through sensitivity, positive predictive value (PPV), accuracy, and the F1 score. Python (3.7.10) with the ‘SciPy’ package (1.6.2) and R (4.1.2, 2021; The R Foundation for Statistical Computing; Vienna, Austria) was used for statistical calculations [19].

## 3. Results

### 3.1. Performance for Information Extraction of 10 Gastric Disease

Table 1 shows the demographics of the 2000 train and 1000 test datasets, there was no significant difference between the train and test datasets. The developed pipeline perfectly processed 94.4% (944/1000) of EGD reports for extracting all the variables in the test dataset. The extracted information on the presence and extent of the five types of gastritis was assessed based on sensitivity, PPV, accuracy, and F1 score (Table 2). Overall performance metrics ranged from 0.966 to 0.996.

Table 3 shows the information extraction performance of NLP for gastric ulcers, polyps, submucosal tumors, dysplasia, and cancer. The metrics for the presence of gastric ulcers was ≥0.977, with a range from 0.952 to 1.000 for location and from 0.956 to 1.000 for the stage. The accuracy of the extraction of ulcer size was 0.999. For the extraction of information on the neoplastic disease, the processing was as close to perfect with a metric of >0.990 for presence, location, and size. Overall performance metrics ranged from 0.975 to 0.999.

### 3.2. Demographics of Gastric Diseases Based on the 10-Year EGD Data

The NLP pipeline was applied to the 2010–2019 EGD reports (*n* = 248,699) from the SNUH Gangnam Center to investigate the prevalence of 10 gastric diseases by sex and age group (Table 4). Among the 10 gastric diseases, the prevalence of three gastric diseases was significantly higher in women than in men: superficial gastritis (17.17% vs. 14.6%), follicular gastritis (0.62% vs. 0.23%), and gastric polyp (7.42% vs. 4.44%). Regarding prevalence by age group, 8 of the 10 gastric diseases showed a positive relationship with age. Atrophic gastritis showed the highest prevalence (83.05%) in patients in their 80s. The prevalence of superficial gastritis and follicular gastritis showed an increasing trend until the patients were in their 40s (18.53%) and 20s (1.53%), respectively, and then decreased with age.

Figure 4 shows the information on the extent and location of the 10 gastric diseases. Atrophic gastritis and intestinal metaplasia showed similar proportions of >60% at an extent of 3. Superficial gastritis, erosive gastritis, and follicular gastritis showed the highest proportions at extent 1. The antrum (i.e., the lower part of the stomach) was the most frequent site where gastric ulcers, gastric dysplasia, and gastric cancer were detected. Gastric polyps and gastric cancer were observed in the body of the stomach at high rates. The gastric submucosal tumor was the only and most frequently found disease in the fundus.

## 4. Discussion

The EGD report and linked pathology report contain details of the gastrointestinal mucosal status observed during the procedure. However, as the reports are usually written in an unstructured narrative format, it requires laborious and expensive manual reviews, which is a challenging approach in the face of the ever-growing annual volume of EGD reports in Korea. In this study, we developed a rule-based NLP pipeline that specializes in the extraction of clinical information from EGD reports written in English, Korean, or both. The pipeline extracted detailed information, such as diagnosis, extent, location, and stage, for 10 gastric diseases with a feasible performance. In addition, we applied the NLP pipeline to the 10-year EGD data and demonstrated the utility of the pipeline for extracting information from the large-scale dataset.

To the best of our knowledge, the NLP pipeline developed in this study is the first pipeline to extract gastric disease information from EGD reports. There have been many efforts to extract clinical information using the NLP technique from gastrointestinal reports. Most of these, however, have been focused on information extraction from colonoscopy reports written in English and on extracting neoplastic disease information [4,9,10,11,12,13,14,15]. In contrast, our NLP pipeline can capture the clinical information of 10 gastric diseases, including neoplastic diseases, from reports written in English, Korean, or both languages. We developed our NLP pipeline in the absence of clinical NLP tools to process the Korean language, and without gastrointestinal endoscopy-specific terms in language systems, such as the Unified Medical Language System and Systematized Nomenclature of Medicine-Clinical Terms, to process EGD reports containing or written in Korean. We believe that our NLP pipeline and NLP dictionary can be considered a touchstone for similar studies.

Most importantly, the pipeline is unique in capturing location and extent information from EGD reports, which are essential for estimating the severity of gastritis, unless an exact classification grade such as the Kimura Takemoto or Sydney classification grades are given. Since a manual review of endoscopic description was the only means to determine the severity, we developed the pipeline to capture location as well as presence information. Given the sheer number of EGD reports generated and collected daily since the implementation of the electronic record system, our pipeline can provide automated extraction of such information and minimize the burden of manual review.

This study has some limitations. The pipeline may not be able to properly process reports with incorrect or inconsistent documentation format. Similarly, EGD reports for institutions or centers with different documentation formats cannot be processed as the pipeline was developed with the dataset retrieved from a single medical center. In addition, since the NLP pipeline in this study was based on rules generated using the sampled development dataset, manual rule updates are required for changes in terminology and report format. To overcome these limitations, clinical NLP systems based on machine learning or deep learning approaches can be applied. Embedding, a deep learning-based vectorization method, learns the relationship between terms to construct semantic similarity and assigns numerically close numbers to terms with high similarity. With this approach, even rare or newly added terms can be approximated to an appropriate value representing the meaning of the term [6,7,8,20,21,22,23]. One of our ongoing works is to apply pre-trained deep learning NLP models, such as BioBERT and recurrent neural network-based models, to extract clinical information from reports. However, the rule-based method we used in this study has the advantage of being straightforward and transparent, allowing clinicians to easily interpret the results, unlike deep learning-based models that provide difficult to interpret the results because they operate as a black box. Furthermore, rule-based methods are easy to modify as medical knowledge changes or updates. Finally, the EGD report contains subjective observations of the individual examiner, which may undermine the reliability of the extracted information. However, information extraction from EGD reports is thought to be the most accurate approach, and in terms of reflecting real-world data, extracting information from procedural notes is still a better approach than analyzing administrative codes such as the International Classification of Diseases codes.

## 5. Conclusions

In conclusion, we developed an NLP pipeline to extract gastric disease information from EGD reports. The pipeline processed the test dataset with high performance. The NLP-derived information from 10-years of EGD big data could be used to study the prevalence of gastric diseases and frequently observed extent and locations. Incorporation of this pipeline into clinical practice may facilitate the quality tracking of EGD procedures cost-effectively and identify unknown relationships between gastric disease and various other conditions.

## Figures and Tables

**Figure 1 jcm-11-02967-f001:**
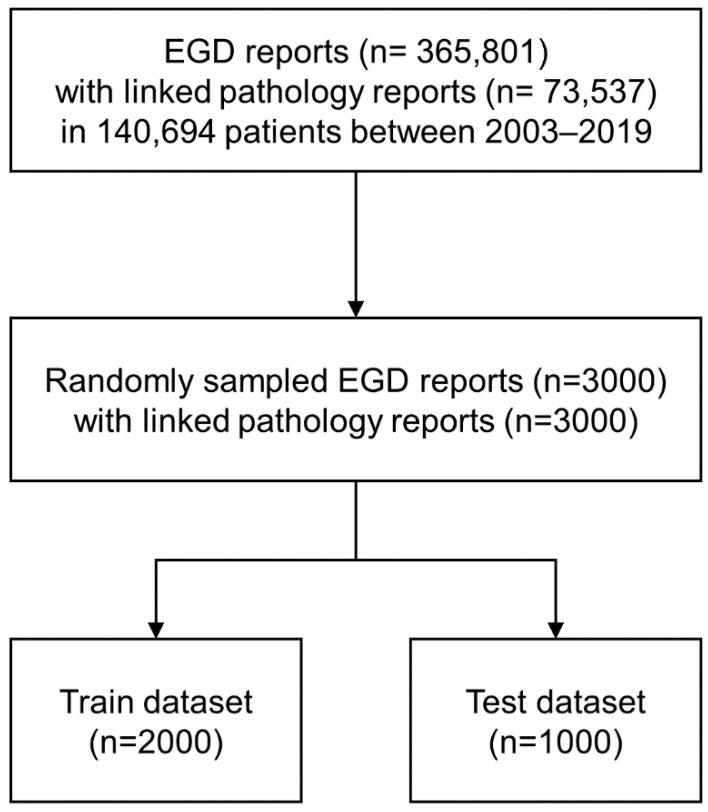
Data flow chart of the study. EGD: esophagogastroduodenoscopy.

**Figure 2 jcm-11-02967-f002:**
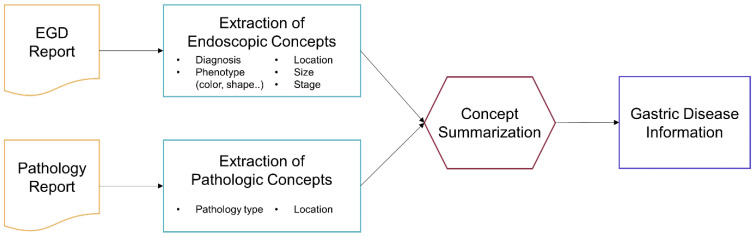
Process of information extraction using the NLP pipeline from EGD and pathology reports. NLP: natural language processing; EGD: esophagogastroduodenoscopy.

**Figure 3 jcm-11-02967-f003:**
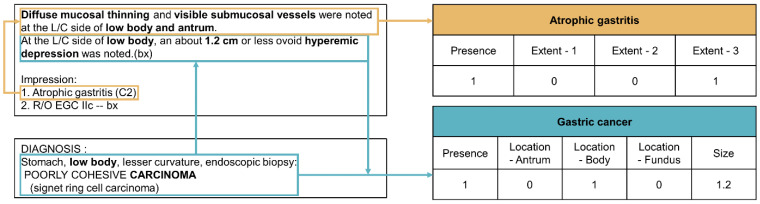
Example of extraction and summarization process of the NLP pipeline. Extent—1: antrum only; Extent—2: body/fundus only; Extent—3: antrum and body/fundus; NLP: natural language processing.

**Figure 4 jcm-11-02967-f004:**
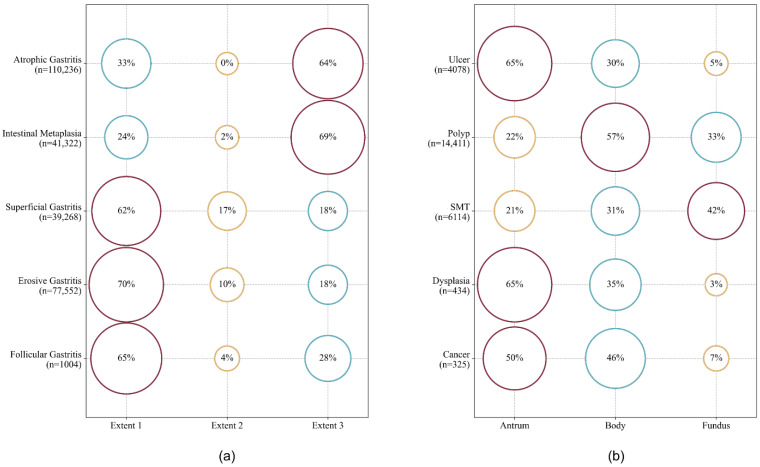
Extents and locations of gastritis (**a**) and other gastric diseases (**b**), respectively, between 2010 and 2019 (*n* = 248,966). The sum of percentages of each disease may be lower or more than 100% due to the reports with unspecified locations and gastric disease at multiple locations in the stomach, respectively. Dysplasia includes tubular adenoma with low and high-grade dysplasia. Cancer includes carcinoma, neuroendocrine tumor, and lymphoma with mucosa-associated lymphoid tissue. Extent 1: antrum only; Extent 2: body/fundus only; Extent 3: antrum and body/fundus; SMT: submucosal tumor.

**Table 1 jcm-11-02967-t001:** Demographics of the train dataset (*n* = 2000) and test dataset (*n* = 1000).

Variables		Train Dataset (*n* = 2000)	Test Dataset (*n* = 1000)	*p*-Value
Age, *n* (%)				0.548
	<30	48 (2.4)	19 (1.9)	
	30–49	761 (38.0)	395 (39.5)	
	50–69	1091 (54.6)	529 (52.9)	
	≥70	100 (5.0)	57 (5.7)	
Sex, *n* (%)				0.643
	Male	1229 (61.4)	605 (60.5)	
	Female	771 (38.6)	395 (39.5)	
Chronic gastritis, *n* (%)				
	Atrophic gastritis	880 (44.0)	415 (41.5)	0.206
	Intestinal metaplasia	491 (24.6)	223 (22.3)	0.187
	Superficial gastritis	225 (11.2)	97 (9.7)	0.228
	Erosive gastritis	1195 (59.8)	597 (59.7)	0.989
	Follicular gastritis	5 (0.2)	5 (0.5)	0.433
Other gastric diseases, *n* (%)				
	Ulcer	163 (8.2)	86 (8.6)	0.726
	Polyp	404 (20.2)	224 (22.4)	0.177
	SMT	80 (4.0)	36 (3.6)	0.663
	Dysplasia *	22 (1.1)	8 (0.8)	0.559
	Cancer ^†^	9 (0.4)	8 (0.8)	0.344

* Dysplasia includes tubular adenoma with low and high-grade dysplasia. ^†^ Cancer includes carcinoma, neuroendocrine tumor, and lymphoma with mucosa-associated lymphoid tissue. SMT: submucosal tumor.

**Table 2 jcm-11-02967-t002:** Performance of information extraction for gastritis using the NLP pipeline.

Variables			Sensitivity	PPV	Accuracy	F1-Score
Atrophic Gastritis						
	Presence		0.993	1.000	0.997	0.996
	Extent *					
		1	0.952	1.000	0.993	0.945
		2	1.000	0.800	0.999	0.889
		3	0.992	0.988	0.995	0.990
Intestinal Metaplasia						
	Presence		1.000	1.000	1.000	1.000
	Extent *					
		1	0.959	0.973	0.995	0.966
		2	1.000	1.000	1.000	1.000
		3	0.985	0.978	0.995	0.981
Superficial Gastritis						
	Presence		0.979	0.990	0.997	0.984
	Extent *					
		1	0.984	1.000	0.999	0.992
		2	0.885	1.000	0.997	0.939
		3	1.000	0.900	0.999	0.974
Erosive Gastritis						
	Presence		0.990	0.998	0.993	0.994
	Extent *					
		1	0.963	1.000	0.984	0.981
		2	0.929	0.988	0.993	0.958
		3	0.986	0.862	0.988	0.920
Follicular Gastritis						
	Presence		1.000	1.000	1.000	1.000
	Extent *					
		1	0.750	1.000	0.999	0.857
		2	N/A	N/A	1.000	N/A
		3	1.000	1.000	1.000	1.000
Overall			0.966	0.972	0.996	0.967

* Extent 1 means gastritis at the antrum only, extent 2 means gastritis at the body or fundus, and extent 3 means gastritis at the body or fundus as well as at the antrum. NLP: natural language processing. PPV: positive predictive value. N/A: not available.

**Table 3 jcm-11-02967-t003:** Performance of information extraction for gastric ulcers, polypoid lesions, and neoplastic diseases using the NLP pipeline.

Variables			Sensitivity	PPV	Accuracy	F1-Score
Ulcer						
	Presence		0.988	0.977	0.997	0.983
	Location					
		Antrum	0.956	0.985	0.996	0.970
		Body	1.000	0.952	0.999	0.976
		Fundus	1.000	1.000	1.000	1.000
	Stages					
		Active	1.000	1.000	1.000	1.000
		Healing	1.000	1.000	1.000	1.000
		Scar	0.977	0.956	0.997	0.966
	Size		N/A	N/A	0.999	N/A
Polyp						
	Presence		0.991	1.000	0.998	0.996
	Location					
		Antrum	1.000	0.964	0.997	0.982
		Body	0.991	0.973	0.996	0.982
		Fundus	0.983	1.000	0.999	0.991
	Size		N/A	N/A	1.000	N/A
SMT						
	Presence		0.972	0.972	0.998	0.972
	Location					
		Antrum	0.750	0.818	0.995	0.783
		Body	1.000	1.000	1.000	1.000
		Fundus	0.833	1.000	0.998	0.909
	Size		N/A	N/A	0.999	N/A
Dysplasia *						
	Presence		1.000	1.000	1.000	1.000
	Location					
		Antrum	1.000	1.000	1.000	1.000
		Body	1.000	1.000	1.000	1.000
		Fundus	N/A	N/A	1.000	N/A
	Size		N/A	N/A	0.999	N/A
Cancer ^†^						
	Presence		1.000	1.000	1.000	1.000
	Location					
		Antrum	1.000	1.000	1.000	1.000
		Body	1.000	1.000	1.000	1.000
		Fundus	1.000	1.000	1.000	1.000
	Size		N/A	N/A	1.000	N/A
Overall			0.975	0.982	0.999	0.978

* Dysplasia includes tubular adenoma with low and high-grade dysplasia. ^†^ Cancer includes carcinoma, neuroendocrine tumor, and lymphoma with mucosa-associated lymphoid tissue. NLP: natural language processing. PPV: positive predictive value. SMT: submucosal tumor. N/A: not available.

**Table 4 jcm-11-02967-t004:** Prevalence of gastritis, gastric ulcers, polypoid lesions, and neoplastic diseases by sex and age between 2010 and 2019 (*n* = 248,966).

		Gastritis					Gastric Ulcer, Polypoid Lesions, and Neoplastic Diseases				
Variables		Atrophic Gastritis	Intestinal Metaplasia	Superficial Gastritis	Erosive Gastritis	Follicular Gastritis	Ulcer	Polyp	SMT	Dysplasia *	Cancer ^†^
Sex, *n* (%)											
	Male, *n* = 136,184	68,719 (50.49)	29,081 (21.36)	19,906 (14.62)	47,792 (35.11)	308 (0.23)	3002 (2.21)	6050 (4.44)	3050 (2.24)	324 (0.24)	230 (0.17)
	Female, *n* = 112,782	41,517 (36.82)	12,241 (10.86)	19,362 (17.17)	29,760 (26.39)	696 (0.62)	1076 (0.95)	8361 (7.42)	3064 (2.72)	110 (0.10)	95 (0.08)
Age group, *n* (%)											
	18–19, *n* = 284	2 (0.70)	1 (0.35)	27 (9.51)	45 (15.84)	3 (1.06)	0 (0.00)	5 (1.76)	4 (1.41)	0 (0.00)	0 (0.00)
	20–29, *n* = 7888	276 (3.50)	38 (0.48)	1179 (14.95)	1425 (18.06)	121 (1.53)	38 (0.48)	320 (4.06)	74 (0.94)	0 (0.00)	4 (0.05)
	30–39, *n* = 32,028	3455 (10.79)	619 (1.93)	5904 (18.43)	7440 (23.23)	358 (1.12)	224 (0.70)	1874 (5.85)	366 (1.14)	4 (0.01)	11 (0.03)
	40–49, *n* = 71,049	21,996 (30.96)	5999 (8.44)	13,164 (18.53)	21,012 (29.57)	314 (0.44)	794 (1.12)	4329 (6.09)	1215 (1.71)	42 (0.06)	52 (0.07)
	50–59, *n* = 85,248	46,125 (54.11)	16,735 (19.63)	13,134 (15.41)	29,084 (34.12)	176 (0.21)	1572 (1.84)	4626 (5.43)	2243 (2.63)	152 (0.18)	119 (0.14)
	60–69, *n* = 39,513	27,937 (70.70)	12,386 (31.35)	4766 (12.06)	14,064 (35.59)	31 (0.08)	996 (2.52)	2366 (5.99)	1560 (3.95)	155 (0.39)	88 (0.22)
	70–79, *n* = 11,841	9519 (80.39)	4998 (42.21)	996 (8.41)	4093 (34.57)	1 (0.01)	392 (3.31)	798 (6.74)	586 (4.95)	63 (0.53)	49 (0.41)
	≥80, *n* = 1115	926 (83.05)	546 (48.97)	98 (8.79)	389 (34.89)	0 (0.00)	62 (5.56)	93 (8.34)	66 (5.92)	18 (1.61)	2 (0.18)

* Dysplasia includes tubular adenoma with low and high-grade dysplasia. ^†^ Cancer includes carcinoma, neuroendocrine tumor, and lymphoma with mucosa-associated lymphoid tissue. SMT: submucosal tumor.

## Data Availability

Not applicable.

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
