# Peer review of "Natural Language Processing for Information Extraction of Gastric Diseases and Its Application in Large-Scale Clinical Research"

_jcm, 2022, doi:10.3390/jcm11112967_

Round 1

Reviewer 1 Report

Interesting study looking at extracting.  I think the article was very interesting.

Data with keywords for endoscopically determined G.I. abnormalities. Could help with the pooling of data and data analysis for a variety of studies. 

I would like a bit more description about the specifics in which the others have both ways to support in similar methodologies used in the past and how they envisioned this to be useful in the future, otherwise I think a well written article and after such comments/clarifications should be appropriate. 

Author Response

Responses to Reviewer #1

Comment.

I would like a bit more description about the specifics in which the others have both ways to support in similar methodologies used in the past and how they envisioned this to be useful in the future, otherwise I think a well written article and after such comments/clarifications should be appropriate. 

Response.

Thank you for your comment. We agree with your point and think that your suggestion will make our manuscript more understandable.

Thanks to the reviewer’s suggestion, we have added and mentioned recent references to studies that have conducted research similar to ours.

Manuscript

[Introduction]

Do et al. used the NLP system to gather patterns of metastatic spread from radiology reports in a large patient cohort [9]. Bae et al. developed an NLP system to extract information related to neoplasm from colonoscopy reports, and evaluated the quality benchmarks of endoscopists [10].

[Reference]

  1. Do RK, Lupton K, Causa Andrieu PI, Luthra A, Taya M, Batch K, Nguyen H, Rahurkar P, Gazit L, Nicholas K, Fong CJ. Patterns of Metastatic Disease in Patients with Cancer Derived from Natural Language Processing of Structured CT Radiology Reports over a 10-year Period. Radiology. 2021;301(1):115-22 doi: 10.1148/radiol.2021210043.

10. Bae JH, Han HW, Yang SY, Song G, Sa S, Chung GE, Seo JY, Jin EH, Kim H, An D. Natural Language Processing for Assessing Quality Indicators in Free-Text Colonoscopy and Pathology Reports: Development and Usability Study. JMIR Medical Informatics. 2022;10(4):e35257 doi: 10.2196/35257.

Reviewer 2 Report

I would like to congratulate the authors on developing such useful tool for analysing endoscopy reports. The manuscript is very interesting and the findings are practical and relevant. I have only few issues that need to be resolved:

1. Table 2 is doubled, but there is no Table 3.

2. Even though Figure 4 is very informative and clear (and it should stay), I would like to see in the main manuscript data from Supp. Table 2, as this is the factual main result of the study - that the developed algorithm allows for big data analysis.

3. I understand the terminology used in the manuscript, i.e. recall, precision, accuracy and F1, as I worked with similar system before. However I would strongly suggest adding or changing the terms to more familiar for the medical public, i.e. sensitivity and specificity.

4. Even though I know what the authors wanted to prove with the adenoma association study (how can NLP pipeline be used for research purpose), I do not feel that this part is needed in this manuscript and feels outside of the scope. Instead, I would suggest showing more data on the whole dataset analyzed (see point 2).

5. Discussion does not include considerations on applying this process outside of Korean language. Is it universal? Does it have potential to work in other languages? It is also not clear to me, whether NLP works for English language or is it just working when some terms are english-derived. Please elaborate.

Author Response

Responses to Reviewer #2

Comment 1.

Table 2 is doubled, but there is no Table 3.

Response 1.

Thank you for pointing out the mistake, and we regret not showing Table 3.

Thanks to the reviewer’s comment, we replaced the duplicated table (Table 2) with the right one (Table 3).

Manuscript

[Results]

Table 3. Performance of information extraction for gastric ulcers, polypoid lesions, and neoplastic diseases using the NLP pipeline.

Variables

Sensitivity

PPV

Accuracy

F1-score

Ulcer

Presence

.988

.977

.997

.983

Location

Antrum

.956

.985

.996

.970

Body

1.000

.952

.999

.976

Fundus

1.000

1.000

1.000

1.000

Stages

Active

1.000

1.000

1.000

1.000

Healing

1.000

1.000

1.000

1.000

Scar

.977

.956

.997

.966

Size

N/A

N/A

.999

N/A

Polyp

Presence

.991

1.000

.998

.996

Location

Antrum

1.000

.964

.997

.982

Body

.991

.973

.996

.982

Fundus

.983

1.000

.999

.991

Size

N/A

N/A

1.000

N/A

SMT

Presence

.972

.972

.998

.972

Location

Antrum

.750

.818

.995

.783

Body

1.000

1.000

1.000

1.000

Fundus

.833

1.000

.998

.909

Size

N/A

N/A

.999

N/A

Dysplasia*

Presence

1.000

1.000

1.000

1.000

Location

Antrum

1.000

1.000

1.000

1.000

Body

1.000

1.000

1.000

1.000

Fundus

N/A

N/A

1.000

N/A

Size

N/A

N/A

.999

N/A

Cancer

Presence

1.000

1.000

1.000

1.000

Location

Antrum

1.000

1.000

1.000

1.000

Body

1.000

1.000

1.000

1.000

Fundus

1.000

1.000

1.000

1.000

Size

N/A

N/A

1.000

N/A

Overall

.975

.982

.999

.978

*Dysplasia includes tubular adenoma with low-grade dysplasia.

Cancer includes carcinoma, neuroendocrine tumor, and lymphoma with mucosa-associated lymphoid tissue.

NLP: natural language processing.

PPV: positive predictive value.

SMT: submucosal tumor.

N/A: not available.

Comment 2.

Even though Figure 4 is very informative and clear (and it should stay), I would like to see in the main manuscript data from Supp. Table 2, as this is the factual main result of the study - that the developed algorithm allows for big data analysis.

Response 2.

Thank you for your suggestion. We agree with your point that moving supplementary table 2 to the manuscript will enhance the main concept of the manuscript. We have moved Supplementary Table 2 to the main manuscript as Table 4.

Comment 3.

I understand the terminology used in the manuscript, i.e. recall, precision, accuracy, and F1, as I worked with similar system before. However I would strongly suggest adding or changing the terms to more familiar for the medical public, i.e. sensitivity and specificity.

Response 3.

Thank you for your comment. We agree with your point that changing performance metrics will make the manuscript more readable. Following your suggestion, we replaced “recall” and “precision” with “sensitivity” and “positive predictive value (PPV)”, respectively, in the manuscript as well as tables.

Comment 4.

Even though I know what the authors wanted to prove with the adenoma association study (how can the NLP pipeline be used for research purpose), I do not feel that this part is needed in this manuscript and feels outside of the scope. Instead, I would suggest showing more data on the whole dataset analyzed (see point 2).

Response 4.

Thank you for your kind suggestion. We agree with your point that removing the association study part from the manuscript will make our manuscript straightforward. Thanks to the reviewer’s comment, we removed the content on the adenoma association study from the manuscript.

Comment 5.

Discussion does not include considerations on applying this process outside of Korean language. Is it universal? Does it have potential to work in other languages? It is also not clear to me, whether NLP works for English language or is it just working when some terms are english-derived. Please elaborate.

Response 5.

Thank you for your kind comment. Thanks to your comment, we realize that there may be some confusion around the NLP pipeline in our manuscript. We added descriptions of the language composition of the reports we used for the development and validation of the NLP pipeline. Furthermore, we altered sentences in the discussion section.

Thanks to the reviewer’s suggestion, we altered the manuscript to deliver the intended information clearly.

Manuscript

[Materials and Methods]

The EGD reports included three types of language forms: Korean, English, and Korean with English terminology. Of the 3000 EGD report in the development dataset, 75 EGD reports were written in English only, and the rest of 2925 EGD reports were written in Korean with English terminology. Reports written in Korean only were not included in the development. Therefore, we needed to create an NLP pipeline that can process multi-language reports written in Korean, English, or both, and build a lexicon of Korean-English medical terms, synonyms, and endoscopic abbreviations (Supplementary Table 1).

[Discussion]

In this study, we developed a rule-based NLP pipeline that specializes in the extraction of clinical information from EGD reports written in English, Korean, or both. 

Reviewer 3 Report

The authors describe a process for extracting medical terms from reviews. On the technique of data management and IT, I do not feel competent to evaluate. On the results, this study seems very interesting.

However, there is one point that bothers me, and that is the evaluation of the frequency of colonic adenomas associated with that of the presence of Helicobacter pylori gastritis. Indeed, this is a technological work of keyword extraction and not an epidemiology work which would seek the association between these two entities in a human population specifically for this subject. All the materials and methods, results and discussion part of the article on this association should be removed.

Author Response

Responses to Reviewer #3

Comment.

However, there is one point that bothers me, and that is the evaluation of the frequency of colonic adenomas associated with that of the presence of Helicobacter pylori gastritis. Indeed, this is a technological work of keyword extraction and not an epidemiology work which would seek the association between these two entities in a human population specifically for this subject. All the materials and methods, results and discussion part of the article on this association should be removed.

Response.

Thank you for your comment. We agree with your point that the association study in the manuscript was out of scope. Thanks to the reviewer’s suggestion, we removed the content of the adenoma association study from the manuscript.

Reviewer 4 Report

I would like to thank the authors for this interesting work. I believe that the study has a good potential. However, there are some points that should be considered in the next version, please.

Major points:

  1. My main critique is that the study should have been positioned in line with the state-of-the-art methods of Natural Language Processing (NLP) using BERT models. In this regard, there are specialized models (e.g. BioBERT), which could be quite applicable in the present work. Anyway, this is something that could be discussed as part of the future work.

  1. Likewise, the introduction and discussion should refer to recent contributions availing of BERT-based models for extracting embeddings from text in clinical or medical documents. For example:

https://doi.org/10.3390/app9183658

https://doi.org/10.5220/0011012800003123

  1. Although the rule-based NLP approach have worked well, I argue that it might be difficult for such approach to deal with inconsistencies or incorrect inputs in the documents, which could be largely unavoidable in free-text notes. That said, please consider this point as part of the limitations as well. It is appreciated that you’ve touched on other possible limitations as well.

Minor issues:

  1. Please follow the journal template for formatting the manuscript.
  2. Also, please proofread the manuscript. Some typos exist, though the article is well-written in general.

Author Response

Responses to Reviewer #4

Comment 1.

My main critique is that the study should have been positioned in line with the state-of-the-art methods of Natural Language Processing (NLP) using BERT models. In this regard, there are specialized models (e.g. BioBERT), which could be quite applicable in the present work. Anyway, this is something that could be discussed as part of the future work.

Response 1.

Thank you for your comment. We agree with your critique of the manuscript. Thanks to your point, we realize that the manuscript did not include current NLP studies. Accepting your suggestion, we mentioned in the manuscript one of our current studies on the development or transferring of pre-trained deep-learning-based models such as BioBERT and biLSTM-CRF. Following the reviewer’s suggestion, we mentioned one of our ongoing studies that transfers a state-of-the-art model for information extraction from reports.

Manuscript

[Discussion]

To overcome these limitations, clinical NLP systems based on machine learning or deep learning approaches can be applied. Embedding, a deep learning-based vectorization method, learns the relationship between terms to construct semantic similarity and assigns numerically close numbers to terms with high similarity. With this approach, even rare or newly added terms can be approximated to an appropriate value representing the meaning of the term [6-8, 20-23]. One of our ongoing works is to apply pre-trained deep learning NLP models, such as BioBERT and recurrent neural network-based models, to extract clinical information from reports.

Comment 2.

Likewise, the introduction and discussion should refer to recent contributions availing of BERT-based models for extracting embeddings from text in clinical or medical documents. For example:

https://doi.org/10.3390/app9183658

https://doi.org/10.5220/0011012800003123

Response 2.

Thank you for your kind suggestion. Thanks to your suggestion, we could add recent and state-of-the-art NLP models as a reference in our manuscript. We have added and mentioned a few recent NLP studies that are using deep learning methods, in the introduction and discussion section of the manuscript.

Manuscript

[Introduction]

For example, extraction of biomedical terms and relations from texts using deep learning-based NLP models, such as BioBERT (Bidirectional Encoder Representation from Transformers for Biomedical Text Mining), recurrent neural network-based models, and Med7, are now increasingly researched [6-8].

[Discussion]

One of our ongoing works is to apply pre-trained deep learning NLP models, such as BioBERT and recurrent neural network-based models, to extract clinical information from reports.

[Reference]

  1. Lee J, Yoon W, Kim S, Kim D, Kim S, So CH, Kang J. BioBERT: a pre-trained biomedical language representation model for biomedical text mining. Bioinformatics. 2020;36(4):1234-40 doi: 10.1093/bioinformatics/btz682.
  2. Yang J, Liu Y, Qian M, Guan C, Yuan X. Information extraction from electronic medical records using multitask recurrent neural network with contextual word embedding. Applied Sciences. 2019;9(18):3658 doi:10.3390/app9183658.
  3. Kormilitzin A, Vaci N, Liu Q, Nevado-Holgado A. Med7: a transferable clinical natural language processing model for electronic health records. Artificial Intelligence in Medicine. 2021;118:102086 doi: 10.1016/j.artmed.2021.102086.

Comment 3.

Although the rule-based NLP approach have worked well, I argue that it might be difficult for such approach to deal with inconsistencies or incorrect inputs in the documents, which could be largely unavoidable in free-text notes. That said, please consider this point as part of the limitations as well. It is appreciated that you’ve touched on other possible limitations as well.

Response 3.

Thank you for your comment. Thanks to your comment, we could include in the manuscript one of the most important limitations of the NLP pipeline that we developed. To address the limitation of “out of distribution” and to enhance the clinical decision support system, we are currently developing a neural network model to measure the confidence scores of given endoscopy reports. We believe that incorporating the model into the NLP pipeline will improve its reliability.

According to the reviewer’s comment, we added and mentioned suggested limitations in the discussion section of our manuscript.

Manuscript

[Discussion]

This study has some limitations. The pipeline may not be able to properly process reports with incorrect or inconsistent documentation format. Similarly, EGD reports for institutions or centers with different documentation formats cannot be processed as the pipeline was developed with the dataset retrieved from a single medical center.

Comment 4.

Please follow the journal templates for the formatting the manuscript.

Response 4.

Thank you for pointing out the mistake. Thanks to your comment, we double-checked the format and revised the manuscript. According to the journal’s guidelines, the conclusions section was added to the manuscript, and we revised it accordingly.

Comment 5.

Please proofread the manuscript. Some typos exist, though the article is well-written in general.

Response 5.

Thank you for pointing out the mistake we could have missed. Thanks to your comment, we could realize that the proofing option in our software was not working properly. Following the reviewer’s comment, we ordered a paid proofreading service.

Round 2

Reviewer 3 Report

ok with modification

Reviewer 4 Report

Thanks, I have no further comments.